# A Climate Change and Sustainability Education Movement: Networks, Open Schooling, and the 'CARE-KNOW-DO' Framework

**Alexandra Okada** [1,*]  **and Peter Gray** [2]

1   Faculty of Wellbeing, Education & Language Studies, The Open University, Milton Keynes MK7 6AA, UK
2   Independent Researcher, Edinburgh, UK
*   Correspondence: ale.okada@open.ac.uk; Tel.: +44-1908-653362

**Abstract:** This study explores the interplay and close cooperation gap between universities, schools, enterprises, policymakers, and wider society for the joint development of actions for CCSE 'Climate Change and Sustainability Education'. We argue that CCSE, as the integration of sustainability and eco-consciousness at all educational levels, should empower learners by providing competences to identify issues and responsible actions to shape a liveable planet for all. Underpinned by the CARE-KNOW-DO theoretical principles, we explore CCSE issues and provide a novel foundation for a new education movement to combine strategies, initiatives, and interventions towards learning ecologies. Findings of our Delphi Study with 27 expert academics, practitioners, entrepreneurs, and policymakers of the UK Green-Forum presents seven recommendations to tackle the CCSE's challenges: 1. Promote flexible real-context curriculum; 2. Foster cross-curricular practices with teachers' training; 3. Establish CCSE definition with benchmarks including skills and qualifications; 4. Enhance learners' agency through the cooperation of stakeholders and organisations; 5. Raise students' passion for nature with a hopeful curriculum; 6. Increase green careers awareness through education, and 7. Implement tangible curriculum through policy-change with equity, diversity and inclusion. We present 60 green-initiatives and 33 green-skills for the CCSE, for empowering students to CARE-KNOW-DO actions towards a sustainable world with green-careers, green-lives, and green-societies.

**Keywords:** climate change; sustainability education; open schooling; CARE–KNOW–DO framework; sustainable educational development; learning ecologies; green-skills; green-initiatives

## 1. Introduction

Societies across the world need Climate Change and Sustainability Education (CCSE) to support responsible development including environmental protection, wellbeing, and the green economy. Putting the right educational frameworks in place is considered fundamental for driving the green skills to ensure healthy life, new careers, and fair societies. Environmental degradation and climate change are major threats of life causing millions of deaths per year as highlighted by the World Health Organisation (WHO) recent report due to polluted air and water, insufficient food, and unsafe shelter [1]. A missing opportunity for people and our planet is that the demand of green jobs has been increasing more than the total qualified people. This gap has been increasing each year as indicated by the global green skills report 2022 [2]. Education has a key role to empower learners with competences for sustainability by educating individuals with enough abilities for green jobs, which are occupations that require expertise. In this article, we argue that governments, business, civil society, and the education sector must work together to foster a scientifically literate society with green competences to secure a green planet shaped by green professionals and green citizens. This involves all levels of education, schools,

universities, and non-formal education providers to empower learners for local and global sustainability over the next five to ten years.

In the UK, for example, the new 'Green Jobs Taskforce' was launched in 2020 to support the creation of two million green jobs and reach 'net zero'. This refers to the transition to a low carbon economy with a high-skill job market capable to keep the balance between the production and removal of the same amount of greenhouse gas from the atmosphere by 2050 [3]. The UK Department for Education's strategy for sustainability was established to support the provision of carbon literacy training for schools, colleges, and universities by 2025. This includes five areas [4]:

1.  Climate education: students will develop better understanding of climate change and greater connection to nature to tackle both the causes and impact of climate change.
2.  Green skills and careers: students will be prepared for career paths to support the net-zero transition, the restoration of biodiversity, and a sustainable future.
3.  Education estate and digital infrastructure: students and their communities will be inspired to live sustainable lives with a green physical environment in and around education settings to promote both their physical and mental wellbeing.
4.  Operations and supply chains: students will be introduced to more sustainable practice for waste prevention, resource efficiency and the circular economy.
5.  International actions: students will be inspired to support international actions to make a difference to children and young people all over the world who are facing extreme weather events.

Sustainability education has a key role in preparing students to identify, analyse and solve problems that affect life on the planet for this and subsequent generations. Wiek et al. [5,6] explain that 'competencies in sustainability' refers to the complexes of knowledge, skills, and attitudes required for individuals to perform successful tasks and problem solving related to real-world sustainability issues. Similarly, Bianchi [7] defines sustainability competences as the interlinked set of knowledge, skills, and attitudes; but including also values for individuals to act and respond effectively to real-world sustainability problems, challenges, and opportunities according to the context.

Educators must prepare students to anticipate risks that threaten sustainability, or more accurately survival, since the term 'sustainability' implies a business-as-usual approach [8]. 'Survival' is an appropriate term to create 'deep awareness' of the depth and seriousness of climate change, biodiversity loss, and other threats [9]. In turn, thinking about survival means thinking about 'whose survival?' and thus about issues of social justice. Educators should therefore design interventions that maximise high-quality education equity, diversity, inclusion, and fairness supported by green values.

In this study, therefore, we define green competences as the effective ability to take 'care' of the planet, to 'know' what to 'do', why and how, to take responsibility for the social, economic, and natural environment, underpinned by values, knowledge, skills, and attitude to anticipate, intervene, respond, and recover considering the interdependent effects of climate change and other factors that affect life for all.

We argue that CCSE, as the integration of sustainability and eco-consciousness with education at all educational levels should empower learners by providing competences to identify issues and responsible actions to shape a liveable planet for everyone, now and in the future. Developing the widest possible range of green competences, starting from an early age, is essential for such active engagement.

A key challenge for national governments, including the UK, has been to increase the uptake of STEAM subjects (science, technology, engineering, arts, mathematics) in schools, thereby enhancing individual employability and national competitiveness [10,11]. However, there has been a lack of focus on green issues within this policy strand, with an underlying assumption that it is enough to produce more STEAM graduates, without any consideration of what they will do once in employment. Whilst it is seen as ethical to intervene in the career choices of young people, there is much less of an appetite to steer Research and Development (R & D) and the STEAM labour market in green directions.

We argue, therefore, that the sustainable educational development, ethical positions, and related green skills to build competences in school are essential to steer the overall direction of socio-economic activity in the direction of survival.

*Competence Frameworks, Partnerships and Key Actors*

Various frameworks have been developed to enhance the green economy at the European level [12,13], lifelong learning [14], responsible citizenship [15], and future-oriented green competences [5–7,16,17], as well as at the UK national level, for example, the Green Skills [3,4,18–21].

There are also many funded projects that are building networks and promoting the adoption of these frameworks [22,23]. One example is the European Union, which has funded more than 15 projects on open schooling to enhance science education towards sustainability in Europe, including the UK (e.g., CONNECT and its CARE-KNOW-DO, open schooling framework, [24]). However, after the end of funding for these projects, it is unclear how their work will be sustained. So far, limited attention has been paid to examining how to adopt and scale up these initiatives to make them sustainable in order to provide the necessary impact for change.

Open Schooling is a term promoted by the European Union [23] that refers to schools as agents of wellbeing. Through cooperation with enterprises, universities, and communities, students work with real-life problems to develop the competences that they need to ensure sustainable life and desirable futures [25]. Open schooling can be implemented through various approaches, enabling students to solve problems or questions supported by families and scientists. In this respect it is similar to for example, citizen-science-based participatory research, consensus conference, cocreation, design thinking, inquiry-mapping, system-oriented dialogue model- and collaborative project-based learning. As the name suggests, open schooling operationalises these concepts in primary and secondary education, through local partnerships.

Whilst educational research has been effective in highlighting the need for a wide-ranging, green competences framework, young people experience frustration with traditional curricula and need new forms of learning to take action using knowledge in real-life contexts. Although many organisations have emerged to promote education for sustainability, there is very little collective work being done and the effects of integrated actions are underexplored. In addition, a key issue is the lack of policy at regional and national levels to support changes.

To bring together national and local net-zero ambitions, it is necessary to create a governance framework, based on the concept of local partnerships, as described in recent studies in the UK [26]. Similarly, in education, to establish CCSE, we argue that further studies are needed to explore how to integrate and operationalise these frameworks in practical ways to solve real-life problems. Cooperation is fundamental to sustaining collaborative learning and knowledge exchange. It should involve science education providers, green enterprises, and the wider community, including families, with a strong emphasis on the participation of education systems.

Rightly or wrongly, accountability has become one of the main drivers in Education [27]. Once accountability systems are in place, governance procedures monitor and evaluate actions and these lead to change. When it comes to CCSE, accountability and governance are missing at the national level. We would suggest that these are essential if we are to see change.

In this study, we explore the interplay and close cooperation between various societal representatives in Education to enhance and ensure green skills interlinked with knowledge, attitudes, and values for all. Young people must be inspired, involved, and affectively and cognitively equipped to contribute to the green economy, green life and green society [13].

We also explore the challenges to the adoption of CCSE in schools. In addition, we introduce the CARE-KNOW-DO framework to examine what is already happening in schools, and the barriers that they face in delivering CCSE. A Delphi study methodology

supports this exploratory study. Finally, we provide a set of recommendations to enable CCSE to be widely adopted, to transform existing practices and to become permanently embedded across the whole education system.

## 2. CARE-KNOW-DO Framework for CCSE

The CARE-KNOW-DO framework [23] has been proposed as a basis for operationalising an open schooling approach, by introducing real-life issues for students to discuss and address, supported by partnerships between schools, universities, and societal representatives. In the context of EU-funded projects such as weSPOT, ENGAGE, and CONNECT; the framework has been used by curriculum developers to promote STEAM career development and a greener curriculum [24].

### 2.1. Principles

CARE is the basis for education in general. Heidegger [28] identifies care as a fundamental constituent of being-in-the-world: something that makes us human. Education has evolved as a way of directing care in certain directions. Originally directed towards religious ideas, education eventually took on a role in determining the life-trajectories of young people. We see that careers are a prominent concern for contributors to the study. However, education has failed to instil care for the planet as a duty. World leaders with narrow, nationalistic concerns are helping to destroy the planet at an alarming rate. We expect such leaders to be able to read and write, and yet we have very low expectations for their ethical approach or their urgent responses to climate change. Politicians are quick to blame education, and particularly teachers, for deficiencies in literacy and numeracy, but have little grasp on the serious ethical values and questions underlying education as currently conceived.

Maslow defined five categories of needs as goal states that motivate people: physiological needs, safety, love and belonging, esteem, and self-actualization [29]. Rather like Maslow's hierarchy of needs [30], therefore, we propose that care, as the first element of the framework, should expand from the individual to the planetary, via a series of steps. Furthermore, each element of the framework reflects a corresponding element of education itself. We see CARE-KNOW-DO as an approach to support teaching and learning and foster independent thinkers [23] from an early age. It is there to augment the traditional way of viewing learning, through the interlinked elements of pedagogy, curriculum, and assessment:

Pedagogy: how things are taught and learned, and the type of relationships that exist between students and educators.

Curriculum: what is taught and learned, not necessarily as explicit, documented content but also as knowledge found by students or implicitly embedded in the system.

Assessment: broadly, any activities measuring progress and attainment, commonly via exams or assignments.

The CARE-KNOW-DO framework looks at education from a wider perspective that includes the needs for society and care for individuals. The following section outlines its underlying principles:

### 2.1.1. CARE (Motivation—Affective Engagement)

Care, from a philosophical perspective, is about where one's attention is focused. We are conscious that there is an Orwellian element to our thinking here. In his novel 1984, Orwell [31] explores the nature of belief and how it can be perverted and steered by the exercise of power (2 + 2 = 5). However, the difference lies in the distinction between open and closed education. In Open Schooling, students experience sufficient transactions to connect what they learn in school with the world outside the classroom, in order to develop judgement, empathy, and care for the future. Care has two common usages, and it is important to harness both if we want to improve climate change education. It means firstly 'looking after' the environment and life around us, as stewards for future generations.

Secondly, it also means something that matters or has value to me, such as, for example, eliminating pollution, poverty, and global warming. Care helps students, in conjunction with experts and society, to develop the ethical values and responsible attitudes that are fundamental to Responsible Research and Innovation (RRI). RRI considers that science education, in conjunction with changes in ethics, governance, gender equality, open access and public engagement, is key for preparing students for lives as responsible citizens and innovative professionals [32–34]. It should help learners to discuss real-life problems and develop research-based, innovative solutions whilst aligning scientific and technological advances with societal needs. This is all relevant for students' affective engagement, and to raise the intrinsic motivation necessary for them to develop agency. It promotes interest and a need to know.

The hierarchy of needs for CARE within our CARE-KNOW-DO framework is therefore as follows:

- I am in education because I care, and others care about me.
- I care about myself and my future.
- I care about the future of others close to me and my local environment.
- I care about distant others and their environments.
- I care about the future of the planet as a global ecosystem.

### 2.1.2. KNOW (Formal, Non-Formal or Informal Knowledge)

In terms of knowing and the curriculum, we need to attend to the process of content selection and the reasoning for the inclusion of certain forms of content rather than others. This intersects with the desire of schools (led by the national curriculum) to get their students into employment, but without the capacity to deal with the massive complexity of the labour market, or students' foregrounds. Learning how to learn will be necessary for jobs and skills that do not yet exist. Skovsmose (2012) describes individuals' foregrounds as being essentially their internal vision of their future possibilities [35]. This vision is unlikely to be clear and straightforward, and for many students is likely to reflect anxiety, disadvantage, mystery, and resentment. This is not a criticism of individual guidance teachers but reflects the disconnection between an education system with a restricted range of outcomes, and the world of work as constantly evolving and highly fragmented. The fast-paced knowledge economy is more complex and demanding. Skills requirements, therefore, reflect an increase of technological complexity and organisational efficiency including hard as well as soft skills such as problem solving, communication, teamwork, and self-management [36]. Open schooling is considered as a possible approach to reducing this fragmentation by helping students connect formal, non-formal and informal knowledge, inside and outside school environments, as well as developing hard and soft skills [24]. It creates opportunities for students to expand their 'knowing' in real-life context and develop skills, as well green competences with others. These mean, for example, independent critical-creative thinking and complex-systemic reasoning supported by evidence-based knowledge discussed with a variety of societal actors—teachers, experts, family, and community members. Knowing what, why, how, where and whom to acquire knowledge is fundamental in a world of fake news. In addition, it contributes to the idea of *Bildung* (a German word meaning something like self-maturation) by preparing learners to increase their agency underpinned by their expanded knowledge built in collaboration with others.

The hierarchy of needs regarding knowledge (KNOW) is therefore as follows:

- I am in education because society wants me to know things.
- I want to know what knowledge can help me in the future.
- We want to share knowledge with others.
- We want to understand why knowledge is powerful.
- We need knowledge that will help to make changes.

### 2.1.3. DO (Actions)

Climate Change requires action by society and government, and education will need to support these actions and changes in the way we live. There have been major advances in pedagogy over the last fifty years, and the curriculum has evolved more or less in step with technological or other advances in society as a whole [37,38]. However, assessment has failed to evolve, is still mainly exam based, and is largely a statistical exercise with little relevance to the real world [39,40]. Climate change requires actions, solutions to the problems of a warming world, and ways of adapting to change [41]. A better way to evaluate educational outcomes would therefore be to look at actions taken or activity resulting from learning. This type of assessment would be based on assessment-in-context [42,43]; for example, students' knowledge-in-action or knowledge-through-action, rather than formal exams and other forms of traditional assessments.

This is not to say that literacy, numeracy, and so on are unimportant, far from it, but these skills mean nothing if they are not used in context. The personal and collective context for these basic skills is far more important than currently acknowledged. This is the case even for students learning functional skills, perhaps in vocational education. The recent addition of entrepreneurship to many vocational courses is a small step in this direction, but a major change in how we see entrepreneurship and the related ethical choices is long overdue.

The hierarchy of needs for action (DO) is therefore as follows:

- Education involves doing things informed with knowledge.
- I think and do things because I want to improve my future.
- We reflect and take action with others so we can collectively improve our futures.
- We take knowledgeable action to change the system.
- We achieve planetary survival with wisdom.

This might seem like an idealised way of looking at education, way beyond what can practically be achieved within the existing system. However, this is really the point. The existing system has brought us to a stage where we see solutions as neat, science-based actions with no real cost in terms of systemic change. As Bendell suggests, we need a new, "deep awareness" of the dangers of climate change [9], which inevitably causes anxiety, fear, and hopelessness. In progressing through the framework, we need to deal with the emotional cost of acknowledging the crisis. Already, school leavers or new graduates are experiencing anxiety caused by high levels of debt, housing difficulties and inflation. Hopelessness and anxiety are born out of powerlessness. A focus on action and activism can lead to pathways for change, which can empower students [44] and could be the antidote to barriers that affect students' learning and futures.

Here, we propose that the framework can be expanded to provide a foundation for a new education movement to support CCSE. Historically, pioneers of educational reform such as Celestin Freinet, Rudolf Steiner and Maria Montessori were regarded as outliers to a mainstream system focused on exam-based assessment, standardisation, and conformity. Sivell [45] sees this collective 'new education movement' as a historical phenomenon, which has nevertheless influenced current thinking, not least via the European Commission and ideas about open schooling.

### 2.2. Practices and the CARE-KNOW-DO Framework

Drawing on the theoretical framework outlined above, CARE–KNOW–DO is a practice methodology for making science more meaningful, engaging, fun, and relevant, enabling students to develop knowledge, skills, and attitude with enjoyment [23,44]. It supports problem- or inquiry-based learning by situating the curriculum content within three integrated phases:

- CARE—refers to students' engagement with real-life problems that matter and motivate them to learn.

- KNOW—refers to students' acquisition of knowledge to understand the problem and discuss solutions.
- DO—refers to students' performance of a science action to develop skills and solve problems using the knowledge learned.

We-CARE: The first stage is mainly informal learning with professionals and family, designed to introduce the challenge around a future-orientated issue, stimulate questions and create a 'need to know' that teachers can harness in the next stage.

We-KNOW: The second stage is formal learning focused on students acquiring the scientific understanding and skills they need to make decisions and take action in the final stage.

We-DO: In this stage, students apply the acquired skills and knowledge to participatory science actions, defining ways to approach the given challenge and minimise its impact.

Some European-funded projects have developed other open schooling frameworks to embed sustainability in schools, for example, the model FEEL-IMAGINE-CREATE-SHARE [46], to promote creativity and critical thinking. CARE-KNOW-DO [23] differs from other frameworks because it is explicitly linked to the school curriculum, although not reliant on it. This is fundamental, firstly, because it deliberately connects abstract science content with life outside formal education to help build understanding. Secondly, without the anchoring to the curriculum that CARE-KNOW-DO provides, it would be hard for practitioners to adopt new practices. Thirdly, it helps students understand how the science they learn is useful for, and connects to, their everyday lives.

### 2.3. Research Questions

To summarise, we are using this framework as a lens for understanding how disparate strategies, initiatives and interventions can be combined and understood in order to create an effective climate change and sustainability education movement. The CARE-KNOW-DO framework in turn provides a set of principles to guide such a movement.

Our Research Questions, therefore, focus on three issues:

- RQ1: What are the challenges of integrating CCSE into the current UK curriculum?
- RQ2: What are the key recommendations for implementing CCSE across disciplines and educational levels?
- RQ3: In what ways could CARE-KNOW-DO be used to integrate CCSE resources and skills into existing school systems?

### 3. Methodology: Delphi Study

The Delphi method is an iterative and anonymous participatory approach used for gathering and evaluating expert-based knowledge to address complex issues. It is also a commonly used tool in public policy to inform decisions supported by a panel of experts in areas such as education, healthcare, and climate change [47,48]. It is considered a useful approach for engaging distinctive stakeholders, representative practitioners and experts in fields related to the issues to be discussed in which individuals give their opinion on a particular subject, then respond to the opinions of others. Expert views are used to produce consensus or technical inputs for policy recommendations supported by scenario building, argumentative discussion, and group decision-making (see Figure 1), with underlying reasons on a specific problem [49].

Delphi technique is considered efficient, inclusive, systematic, and structured and can be implemented through group discussion, iterations, and anonymity. The classical approach broadly comprises six steps: (1) Planning (2) Participant Recruitment (3) Data generation (4) Feedback to Participants (5) Subsequent round of questions (6) Iteration until the goal is reached [49].

However, the literature suggests that this method has some limitations, for example, time consumption, attrition rate and participants' preconceived ideas. To overcome these issues, this study focused on promoting productive online meetings based on participants' availability combined with preliminary data generated with an open questionnaire. Thus,

in this study, data were generated during two online sessions of 45 min each, with 27 participants using: (1) Qualtrics for a brief questionnaire, (2) *ZOOM* for in-depth discussion and (3) *Padlet* online noticeboard for generating brainstorming of resources and references to complement the consensual conference report.

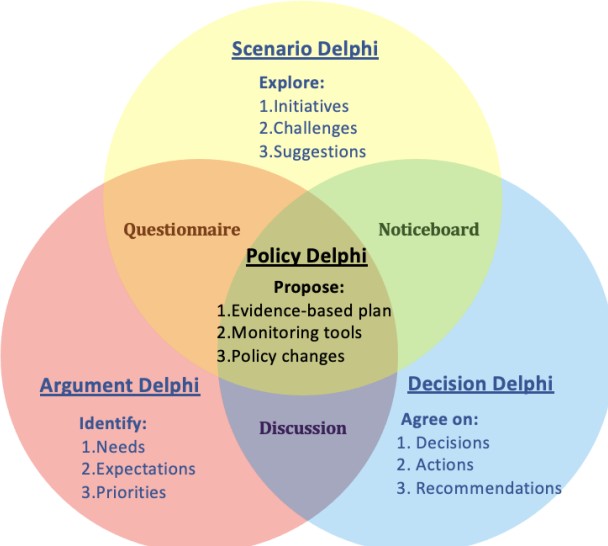

**Figure 1.** Policy recommendations with scenario building, argumentative discussion, and group decision-making (prepared by authors adapted from [48]).

This study, which is part of a research programme on open schooling for sustainable development, was funded by the Open University, with the aim of exploring how projects on sustainability could become sustainable after the end of their grants. In the UK, this Delphi study is related to three EU funded projects: weSPOT, ENGAGE and CONNECT. All these projects focused on RRI—Responsible Research and Innovation with open schooling for Sustainability. This study was developed from June to July 2023 and was approved by the Ethics Committee of the Open University. Anonymised transcriptions and database generated by a questionnaire were analysed using the research evaluation framework developed in CONNECT which in turn is based on open science.

This Delphi study comprised six phases [48,49] described as follows.

### 3.1. Planning

First, participants were selected from a network that was formed during the UK national event on Climate Change in 2022 with 50 members who were interested in further discussions on how to engage, implement and expand Climate Change Education. This network was constituted by academics, policymakers, education directors, consultants, project coordinators, teachers, teacher educators, and programme chairs. They represented various organisations in the UK, for example, universities, schools, science enterprises, educational publishers, science centres, zoos, outdoor environments/parks, business consultancy, research, and policy consultancy.

All members of the green forum received an invitation to attend an online workshop "Education Policy and Climate cHange (EPOCH)—Building networks across boundaries". They were informed that the aim of this event was to enable policymakers to learn from each other, supported by Delphi principles and procedures including iteration, reflection, and anonymity. They were genuinely committed to Sustainability Education and Climate Change and joined the event based on their availability and willingness to volunteer.

Out of 50 members contacted, 27 people signed the consent form to join the Delphi study and completed a form with their interests, occupations, and areas of expertise.

### 3.2. Participants' Profiles

The data provided by participants enabled us to identify a set of common interests and a variety of profiles with different occupations and expertise.

Types of participants' expertise were grouped into four clusters, with a reasonable balance across the clusters, as shown below:

- Cluster 1—Policy and Research: there were six experts in policy research, research on climate education, coordination action and research, innovation funded projects, responsible research and innovation, and consultancy.
- Cluster 2—Curriculum and Teaching: there were eight experts in learning programme on environmental education, curriculum design, primary education and lifelong learning, climate change and environmental education workshops, school initiatives, and state school practices.
- Cluster 3—Teacher Education and Management: there were seven experts in initial teaching education programmes, teacher training, activism and professional practices, and climate action leadership programme.
- Cluster 4—Sustainability Education business: there were six experts on education learning technologies development, educational resources production, multinational publisher in education, and curriculum consultancy.

Their specific professional experience varies from writing course materials, preparing lesson schemes, developing teachers' schemes of work, designing, and delivering workshops for students, supporting local schools on policy and action plans, elaborating policy reports, curriculum planning, developing educational resources and technology, and implementing and evaluating educational programmes, amongst others.

Most participants' interests focused generally on formal education in school and in universities, as well as non-formal learning in science centres, museums, parks, and zoos.

Participants were specifically interested in:

*exploring how to sustain networks on Climate Change and Sustainability Education; managing and delivering learning programmes effectively, scaffolding a fundamental shift in education towards increased awareness with action for sustainable development and green competences, building direct teaching and learning resources that actively support the curriculum, scalability of conservation, environmental and climate education programmes, expanding and integrating existing networks as well as developing new ones, and identifying policy initiatives to make Climate Change Education a reality.*

### 3.3. Initial Data Generation

This Delphi study was designed with three rounds for data generation.

The first round was implemented in two steps. First, participants received information about the common theme, with the Zoom link of the UK Green Forum event that they all attended available online for replay. Second, they received a Qualtrics open questionnaire to answer three questions:

I. What is Climate Change Education?
II. What policy initiatives might be required to make Climate Change Education a reality?
III. What kind of networks, resources and references would support these policy changes?

### 3.4. Feedback to Participants

The second round was implemented through a consensus discussion in Zoom supported by the research coordinator and the Delphi facilitator. Preliminary outcomes related to round 1 were presented by the coordinator who highlighted the challenges presented by current reports on sustainability, climate change and education. Climate change education is crucial to help people understand and address the impacts of the climate crisis and empower young people with the knowledge, skills, values, and attitudes needed to act as agents of change [50]. Sustainability Education is necessary to prepare people

to change their attitudes and behaviour and also develop skills to make informed decisions [50,51]. Education is considered one of our key weapons in the fight against climate change [52]. Teachers must be equipped in every school to deliver world-leading climate change education to empower young people to build a sustainable future.

A summary of the data from the questionnaire was presented to participants to stimulate thinking about the definition of Climate Change Education. Many participants mentioned that, in practice, it means 'embedding 'climate change into the curriculum', 'embedding it within the state school system', 'embedding it in our organisations', 'learning how to minimise the damage to our biosphere', 'lifelong learning for sustainable development', and 'a roadmap for climate action', amongst others.

Regarding the policy initiatives that might be required to make Climate Change Education a reality, participants suggested 'engaging educational providers in knowledge exchange', 'changes to the National Curriculum', 'Climate Change and Sustainability Education needs to be embodied within the National Curriculum' and ' . . . integrated to all aspects of the curriculum', 'making elements of the Sustainability and Climate Change Strategy compulsory and appropriately funded and supporting this'. 'Massive change in how we do 'school'—including more outdoor learning, less content in the curriculum, and a revision of how we present the landscape'; 'Changes in Ofsted requirements, reduction in other pressures—especially at secondary with the focus on exams', 'making it an Ofsted priority; dedicated training for staff', 'Green energy use, climate change education directives akin to the RSE ones', 'curriculum design, commitment to teacher training through the profession at all levels and subjects, whole systems approach', 'climate literacy integrated into the curriculum and assessment', ' . . . including accountability for leadership team/improvement plans within schools'.

Participants agreed that prioritising practical and concrete policy actions with appropriate support, incentives, and funding was the key to success.

'Policy, not strategy, in education that is integrated and holistic . . . needs to support the promotion of action. For educators to support the development of teachers and children there needs to be funding for professional development to raise the understanding of the educators about systems thinking, collaboration and dilemma problem solving. In particular, climate education should not just be through the traditional routes in secondary education or HE.'

'Give courses that have carbon literacy embedded a funding uplift to encourage/incentivise action; funding for estate decarbonisation within education; initiative to 'map' the education estate across a region—locally, regionally, and nationally to see how we can join up that estate for greatest impact on our environment.'

In terms of networks that are necessary to support policy change, participants mentioned initial teacher education networks, multidisciplinary networks with climate and environment experts and curriculum designers to embed research and innovation into the curriculum, a national curriculum that integrates action for climate empowerment, strategic leaders' networks capable of shifting from policy advocacy to policy implementation, with representation from schools academies, NGOs, science centres, and the Government (especially DfE. Ofsted), education, conservation/environmental sectors, and business.

A Padlet was used to collect resources and references suggested by participants (See ORDO data—Padlet open access). Additional comments suggested the need to discuss the topic in depth in groups.

> 'I am not sure teacher/school networks will support change until there is a viable and visible alternative model—which means that policy/activist groups should support these policy changes—ahead of practitioners.'

> 'Which networks bring together the education sector with conservation practitioners and organisations, that also feed into and connect with policy makers?'

> 'There are quite a few organisations now delivering sustainability education but very little collective working—we are geographically spread and need a focus/reason to meet'.

'*Working with experts in the field ESD, climate scientists, and educators and students*'.

### *3.5. Subsequent Round of In-Depth Thematic Discussion*

The third round consisted of an in-depth thematic discussion focused on eight issues that emerged during the interactions, which were introduced and supported by the facilitator to reach a consensual overview.

1.  Rigidity of the Curriculum
2.  Teachers' professional development needs
3.  Key concepts in CCSE
4.  Collaboration—purpose and strategies
5.  Positive vision to empower students
6.  Green skills and careers issues
7.  Governance and Policy change

### *3.6. Iteration till the Goal Is Reached*

The thematic discussion facilitated by collective thinking enabled productive reflection and interaction among participants. Instead of a series of iterations with separate individuals using questionnaires, the approach adopted (Figure 1) was argumentative interaction with decision-making and scenario discussion for policy recommendations. This approach adapted from [48,53] led to more in-depth thoughts and more meaningful results. The conversation was transcribed and is analysed in the following section to reach the findings.

The report on findings was shared through the group discussion list, including the interactive digital interface that was used during all phases, comprising a Padlet for collecting additional resources, comments, and preferences to identify relevance.

## 4. Findings

Figure 2 presents the corpus graphic of the transcribed online discussion with two sessions produced in ZOOM (https://zoom.us/ (accessed on 12 July 2022)) and analysed in NVivo 1.7 (https://www.qsrinternational.com/) (accessed on 12 July 2022).

**Figure 2.** Corpus analysis of the iterative discussion. (Prepared by authors in NVivo).

Climate and Education were the key words most mentioned by participants. These were followed by Curriculum, which was the key concept that emerged in both sessions. In the first session, 'curriculum' emerged together with 'climate educators', whereas in the

second session, curriculum arose together with 'climate education'. In our view, this is indicative of the potential to move from a fragmented approach in which individual educators or organised groups deliver specific activities, to a collective approach or 'movement' with agreed principles and a high level of interaction between players.

Figure 3 presents a summary of the thematic analysis with a description of seven challenges and seven recommendations, detailed in response to our research questions.

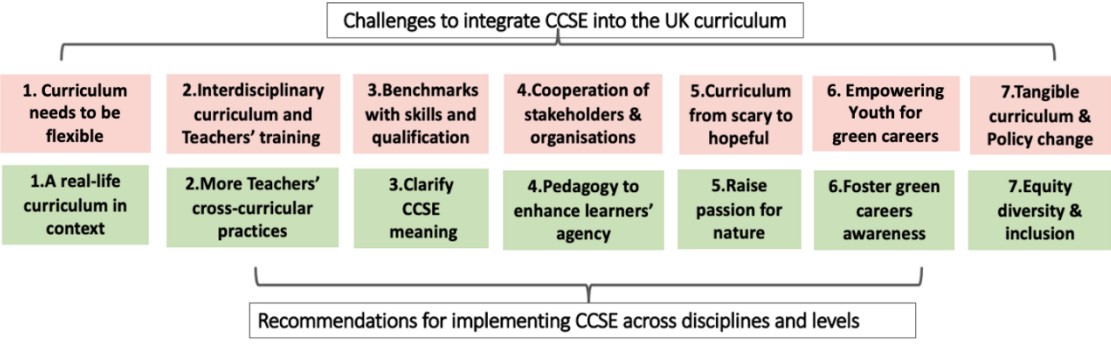

**Figure 3.** Challenges and recommendations to embed CCSE in the curriculum. (Prepared by authors).

*4.1. RQ1. What Are Practitioners' Views on the Challenges of Integrating CCSE into the UK Curriculum?*

4.1.1. Challenge-1: Curriculum Needs to Be Flexible

The first challenge identified was the need for the curriculum to be updated faster in order to align sustainability to real-life issues. Participants mentioned that the UK curriculum is fixed, pre-established, and disconnected from topical issues. Therefore, there is a need to make the curriculum flexible, easy-to-upgrade and contextualised to prepare students for contemporary problems, with real-life questions and topics that are relevant for sustainability.

> *"Can the curriculum accommodate new things, for example, heat pumps? ( . . . )"* (Participant-EPG1)

> *"The conventional curriculum is driven by a process of adding and subtracting topics . . . Living curriculum is driven by an unpredictable flow of events, technological innovations, and cultural change".* (Participant-ENN2)

> *"The heat pumps, although not a new invention, only became a 'hot topic' (sic) due to the need for carbon reduction and the consequent change of gas as a source of domestic heat. The curriculum needs to provide time and space for innovations to come and go, keeping teaching and learning basic principles up to date."* (Participant-EPG1)

4.1.2. Challenge-2: Interdisciplinary Curriculum and Teacher Training

The second challenge indicated was highlighting interdisciplinarity approaches and teacher training. Participants discussed that whilst the scientific aspects of climate change, for example, biology, chemistry, and more specifically ecology and environmental studies are important, there are other issues such as the natural history, the sociology and politics of climate change, which fall under the remit of humanities subjects. Natural, formal, and social sciences should be enhanced through interdisciplinary and transdisciplinary approaches. In addition, participants discussed how teachers could increase their teaching skills and confidence on this topic.

> *"Looking at teachers who are going into primary and secondary and actually how can they take sustainability and climate change and how can they actually do that in every curriculum area and not just the kind of core ones like geography and science that we know are usually how this area is tackled? . . . So it's really starting to look at upskilling those teachers and having them, the confidence especially in primary school, because*

*we're finding if you know they might not have a science background and being able to understand where to go and how to tackle climate change because it is seen as a scientific subject".* (Participant-ECH4)

### 4.1.3. Challenge-3: National Benchmarks, Including Skills, and Qualification

The third challenge centred on the implementation of a national benchmark for green competences, for example, a standardised model describing skills and qualifications to support formative assessment with accountability, learning goals and achievements. In addition, participants commented that this benchmark should be useful for formal and non-formal education, in and beyond schools, and could be implemented in a feasible time frame.

*"There's less and less time and opportunity to get a good climate change sustainability education in there and what's needed . . . things like benchmarks, so career education for example a while back was kind of slipping in schools and it's almost like that sort of opportunity where there needs to be a climate change benchmark that forces schools to do something a little bit more proactive and . . . improve that accountability. Really . . . there's lots of stuff around that talks about broader things and schools going further, and there are examples of schools that will do a lot, but it gets much more difficult, and I do think there's a vehicle needed to kind of push it that little bit more you know."* (Participant-ESW6)

*"Sustainability is not in the curriculum and so therefore you know it's very hard to design fitting qualifications"* (Participant-EBT9)

*"The point about timescales is vital. Climate change is not going to wait for curriculum reform, changes to disciplinary boundaries or changes in governmental philosophy. We are currently seeing an emphasis on growth from the government, which conflicts with long-term solutions. With qualifications that are going to be representative and accessible starting in primary schools and moving through . . . but those processes take a long time, and we don't have a long time . . . "* (Participant-EPG13)

### 4.1.4. Challenge-4: Cooperation between Stakeholders and Organisations

The fourth challenge indicated that scientists, educators, and community members including students and policy makers should collaborate in encouraging the necessary policy changes to make schools agents of well-being, which is the intention of open schooling. Specific aspects of science need to be promoted with more up-to-date topics. In addition, there is something over and above science that might be required to achieve genuine climate awareness, based on a social justice agenda. The lack of power to influence climate change decisions is a major issue for students whose domestic situation is already difficult. However, information and evidence are vital elements in successful decision making and campaigning. The more students are prepared with the skills of using scientific evidence and participatory science, the stronger their decision, intervention and outcomes are likely to be, as acknowledged in the following quote.

*"We do a lot of outreaches with local schools but . . . I think sometimes there can be some elements of climate research that are difficult to make accessible to different age groups, so we would really be happy to explore this and you know sit alongside people and look at the resources and try and think of different areas of research, because for instance I use satellite observations . . . earth observation is a really important part of looking at monitoring climate and understanding how to manage our resources, so little things like this, can we bring them into the into the school grounds for instance even making measurements of surface temperature, you know that surface temperature you [can] measure from space, it's so important for so many different areas of climate research and something like that, if we could do this with like a sensor network in schools or something that might help students think about how they want to interact with the climate and possibly you know maybe there are bigger things that we can build on this."* (Participant-EH15)

Cooperation between schools, universities, and other organisations is also necessary. Schools are reliant on information that percolates through to them via the various designers of curricula and producers of learning materials, including external organisations, for example publishers, research and education centres, and science media companies. Acquiring up-to-date knowledge, as well as real life problems regarding what is actually happening in climate change science or the green economy, can motivate students to be more active and reduces the cognitive dissonance between what they read in textbooks and what they see in the media.

*"I think that's a really key point is the careers and linking with businesses and different organisations because although we aren't, I don't think it's fair to expect the curriculum to be able to change at the fast pace of what science is changing at, but what is changing is organisations and businesses . . . and organisations knowing this, so . . . it's more being able to have the curriculum saying that they need to link with those organisations because they're the ones with the up-to-date data, they're the ones with the up-to-date knowledge and it's actually just ensuring that schools are really connected to those organisations so that they can actually learn from them and bring those that curriculum to life and by engaging with organisations rather than actually asking the curriculum to stay up to date."* (Participant-ECH04)

### 4.1.5. Challenge-5: Curriculum Must Move from Scary to Hopeful

The fifth challenge highlighted by participants refers to the effects of climate change and sustainability issues provoking students' and teachers' discomfort within the curriculum. The issue of climate change as 'scary' and the corresponding desire to 'prepare for a new normal' affects whether we choose to prioritise existing curricula (business as usual) or to accept that building resilience is central. This might involve making time for dialogue and counselling, and in turn giving teachers the tools to deal with social issues outside the curriculum, as in fact they have already had to do in connection with, e.g.,

*"[in] the primary point of view, which is how to just how to position this, not as scary climate change but as a learning sort of a curriculum that's informed by the change that's going on around us. I think especially for primary we need to move away from a scary climate change . . . a scary unsolvable problem to "this is preparing for a new normal, for a life with serious [changes] in a climate crisis . . . "* (Participant-EEE16)

*"We need to look at future [teacher] training, we need to look at all of those different levels, we definitely need to have a hopeful pedagogy and embrace that, because it's not just about the discomfort of the children but it's also the discomfort of the teachers we've got you know we haven't got enough subject knowledge within our teaching staff we haven't got enough knowledge of appropriate pedagogies to deal with these issues as well as the emotional impact and discomfort for our teachers . . . "* (Participant-EVJ20)

The phrase 'a hopeful pedagogy' sums up the need for teachers to deal with the social and emotional aspects of climate change by reducing anxiety and hopelessness, to which young people are increasingly prone.

### 4.1.6. Challenge-6: Empower Youth towards Green Careers

This sixth challenge is a key insight. The effects of climate change are determined far more by major socio-economic decisions than by micro-actions at the domestic scale, useful though these might be. Recent events such as Brexit (the withdrawal of the United Kingdom from the European Union), the invasion of Ukraine, economic crisis and poverty growth, or the scandal over the export of UK plastic waste to Turkey, underline the need for a higher level of climate consciousness at all levels up to world leadership. This can only be achieved if this consciousness becomes embedded in the same way that literacy and numeracy have been embedded in education over the last 150 years or so.

*"I'm doing my bit but actually having the message to young people that choosing a sustainable and ecological and social justice career is not only the right thing to do but*

*will be a successful career for you so that you can imagine yourself having success and this isn't just about doing your bit and switching off the lights and being part of a very small mechanism but it's actually about the individual being successful in their life journey as well but I think it it's framing it in all of these different ways yeah that's really important and it occurred to me while the Expert in the last meeting was talking a couple of weeks ago that uh we can't simply be making kids feel responsible for the whole thing and feel that it's up to them to change their own lifestyles and behaviour, it's about them being responsible for other people's actions.*" (Participant-EVJ20)

### 4.1.7. Challenge-7: Tangible Curriculum with Policy Change

This seventh challenge refers to evidence-based policy. Participants indicated that the UK Government has a strategy but not much in the way of concrete actions. In our opinion, this is not necessarily a bad thing, since the main requirement is for space and time within existing curricular frameworks, rather than adding additional topics and responsibilities. Participants highlighted that providing tangible curriculum achievements supported by real and manageable outputs will enable policy change and impact.

*"Until schools can see a tangible curriculum in place or a tangible curriculum they can't imagine it, it's very hard . . . to take that journey without it in place and I don't think from my limited experience with policy is that policymakers at the moment . . . the department for education is not going to put anything on the table, they want schools . . . they want it to come from within the system and I think that schools are scared for all the accountability reasons so certainly you know whether I'm you know there's conversations (within a large publishing enterprise in education) with all these organisations but at some point i think we need probably a few tangible examples of full curricular [changes] perhaps you know what would it look like to not have all those . . . core curriculum subjects how else would you cover it you know actually making that look real and manageable is, I think, a big part of changing and demonstrating to schools that there are serious organisations who will support and inform a revised curriculum delivery that accounts for the sustainability and learning that we all believe is important.*" (Participant-EEE16)

### 4.2. RQ2. What Are the Key Recommendations for Implementing CCSE across Disciplines and Educational Levels?

Based on the notes taken during the national online event on climate change and participants' comments provided in a ZOOM meeting (oral discussion and written messages in the chat), seven recommendations emerged:

### 4.2.1. Recommendation 1—Encourage the Development of a Flexible Real-Life Contextualised Curriculum for Educators

The first recommendation to support the curriculum to move faster is a real-life contextualised curriculum. Climate change is the ultimate real-life scenario, and one of the key findings from the study is that all participants support a move away from abstraction, especially in science subjects, and a move towards real-life topics and scenarios. This does not preclude the use of theory and abstraction but provides a context for their deployment as tools for problem solving. Teachers need the flexibility and education on how to include topical issues and real-life problem scenario.

*'The current curriculum has lots of scope for this, but teachers just don't know how, don't have the time to find these novel examples and develop resources around them'.* (Participant-E19ZZ)

*"Teachers have been using real life examples in their lessons, which is great because that's something I talked a lot about for the past five years that we need to embed across all curriculum areas"* (Participant-ELD00)

### 4.2.2. Recommendation 2—Increase Teachers' Cross-Curricular Practices

The second recommendation towards an interdisciplinary curriculum for all ages (5–25) and educational levels (from primary school to higher education), supported by teacher professional development, is to encourage and prepare teachers to implement cross-curricular practices. This is an important point, particularly in view of the emergence of phenomenon-based education or the 'deliberative curriculum' [53]. This involves students not only in discussions and investigations around pre-determined topics, but in choosing the topics themselves and debating the justifications for their choices, thus reinforcing their own agency and ability to care and to do. This leads into the next issue from the data, 'students' passion', as in the following extracts.

*"As was said in the national forum, the curriculum across different jurisdictions world-wide reflects or includes environmental and sustainability education as one of the most articulated cross-curricular themes"* (Participant-E12MR)

### 4.2.3. Recommendation 3—Clarify the Meaning of Climate Change and Sustainability Education (CCSE)

The third recommendation to support national benchmarks with skills and qualifications is to clarify the meaning of Climate Change and Sustainability Education. There is still a lot of confusion over what influences climate and what does not, and how education could promote climate change and sustainability. The following quotes suggest that Sustainability and Climate change are not well integrated into schools. There is some evidence that these can be accommodated in e.g., the forthcoming natural history GCSE or Environmental Science A level and embedded at each Key stage together with careers advice.

*"You can't expect year old pupils to understand how to work out a carbon footprint just putting it out the air you know, and I've got that in year two and year five in my [scheme] to work so until we get it in to children what climate change is about what sustainability is about and essentially why we need to know about it you know what effect is it having until they know that we're going nowhere"* (Participant-AE02)

*"It's just not evident at all you know, sustainability isn't there and so it's a very difficult position to be in where you're trying to design a curriculum, and you don't really have a level playing field to start from because how do we start talking about you know sustainability in terms of product design even you know global implications inequalities those types of things outside of the sustainability window ... and um you've got some misunderstanding in terms of what is a carbon footprint what are recycling processes, what you know about how can I be sustainable"* (Participant-EBT9)

### 4.2.4. Recommendation 4—Promote New Pedagogies through Teacher Professional Development to Enhance Students' Agency with Real-Life Issues

The fourth recommendation is to bring together stakeholders and organisations to empower students with the green competences needed to influence climate change decisions, interventions, anticipation, and responsiveness. It is necessary to promote new pedagogies through teachers' professional development to enhance students' agency, in order that they can tackle real-life issues.

Student and teacher agency is vital in supporting climate change actions (DO) rather than just the CARE-KNOW aspects. Current UK policies combine micro-management of content and assessment with agency at school management level, driven by the need to satisfy league table and the like. The key to embedding CCSE and the agency that this requires is to bring Ofsted on board. This should be achievable in view of the Government's current strategy in this area [4], which mentions CPD specifically and links it to other initiatives such as the National Climate Education Action Plan [54]. Reading the strategy document and the Action Plan, we are struck by the convergence in ideas between these documents and the views of our participants. The challenge is to integrate the practices,

real-life scenarios and student agency needed for CCSE into the strategic framework and to make them 'stick', irrespective of short-term policy changes.

> *"The forum was very useful to discuss today's agenda that requires a different pedagogical repertoire. "Could pedagogy actually be an answer if curriculum is imposed on us? How can we interact with that as teachers and leaders? Does that give us some agency and autonomy to be able to make a bigger difference? Can we make a bigger difference to the lives of the children, learners in front of us in relation to sustainability and climate change education?"* (Participant-EALH01)

The question of teacher professional development (TPD) and initial teacher education (ITE) is complex. Building new skills into ITE is as slow as curriculum development and often slower, whilst new forms of TPD require buy-in from school leaders and education authorities, despite being encouraged in policy documents. TPD is also problematic in terms of adding to workload and is not universally popular amongst teachers.

A possible step forward might be to collectively implement some form of Green Charter or quality mark, in which schools are assessed on their deployment of CCSE at a whole school level.

A key factor debated from the previous discussion forum was that:

> *"Change needs to be much more than simply providing materials with curriculum links that can be downloaded as lesson plans . . . off the shelf. If we're going to provide educators with support to teach links to sustainability and climate change well then, they need to know how to do that so that brings in a bigger question of professional development for experienced, early career and trainee teachers."* (Participant-EAAR01)

### 4.2.5. Recommendation 5—Raise Students' Passion for Nature

The fifth recommendation towards moving the curriculum from 'scary' to 'hopeful' is to create opportunities for raising students' passion for nature, including positive attitudes, values integrated to knowledge and skills. This is also recognised in current policy documents, mentioned above, which acknowledge the need for outdoor experiences to build students' concern for the natural environment. Despite the possible introduction of a natural history GCSE in 2025, however, students do not have a mechanism for accrediting informal learning of this type, and external initiatives, such as the awarding of digital open badges (https://support.mozilla.org/en-US/kb/why-open-badges) accessed on 15 November 2022, might be necessary to fully reflect the diversity of such learning experiences.

> *"'We want learners who are passionate about the natural world and this will not come from curriculum content alone. It embraces Environmental Education, time in the outdoors and values driven conversations.' One of the participants mentioned that it's not possible to [convert] these different elements to a love of learning for the environment in which we live".* (Participant-EALH01)

> *"Another participant said that 'It's the much needed wow factor of a new stunning landscape and experience beyond the usual school or home environment that engages curiosity, reinforces knowledge and ultimately inspires passion and commitment to care for our environment'"* (Participant-EAMC01)

> *"It was said that "almost any subject could be provided in outdoor spaces so you'll see in our education estates and some of our innovations we're looking at outdoor learning pods, how you can provide the right environment to use the technology outside while also bringing in outdoor elements"* (Participant-EAJD01)

### 4.2.6. Recommendation 6—Foster Students' Green Careers Awareness

The sixth recommendation to promote education for green careers is to foster students' green careers awareness. This highlights the difficulties already inherent in steering students—including disadvantaged ones from the most vulnerable areas—highly affected by climate change and climate injustice—towards particular careers through advice based

on strategic policy. The area of 'green jobs' is highly dependent on economic and industrial policy decisions, which are inherently changeable over short timescales, as we are currently seeing. The installation of heat pumps or solar panels, additional insulation or the manufacture of electric vehicles might well create 'green jobs', but it is also likely that there will be skills shortages in other areas such as agriculture. Our interpretation here is that career guidance as currently configured is not really much help to students, who would benefit more from the combination of developing green competences useful for all careers and obtaining generic personal development than from advice based on changeable trends.

> *"It was mentioned that "Actually, the traditional routes we would have thought for career guidance like through careers work and work experience students weren't finding out as much about green careers."* (Participant-EALD01)

> *"As with issues such as safeguarding, benchmarks and norms are important to scaffold the growth and embedding of CCSE within the existing system. Developing a green competence framework is part of this, and it is likely that some of the work will involve formal structures and perhaps legislation in order for this to happen".* (Participant-EPG01)

> *"Qualification and decision-making skills are vital. How can we make Carbon Literacy a 'functional skill' beyond schools, and for all?"* (Participant-EO001)

### 4.2.7. Recommendation 7—Support Equity, Diversity, and Inclusion

To consider a tangible curriculum that is enhanced by, and enhances, policy change, the seventh recommendation is to promote equity, diversity, and inclusion. Sadly, for many disadvantaged learners, if this is not part of their school curriculum, they are likely to miss out on climate education. Every child should have the opportunity to experience a night under the stars in an awe-inspiring natural environment.

> *"I agree with the forum participant that we must have the means to enable people of all ages and backgrounds to be able to learn. This means not just embedding green issues into the curriculum and creating new qualifications but making it easier and affordable for all people to study"* (Participant EAEC-001)

### 4.3. RQ3. In What Ways Could CARE-KNOW-DO Be Used to Integrate CCSE Resources and Green Skills Frameworks?

To respond to this question, data generated during the synchronous and asynchronous interaction with participants about current resources, recent reports and adopted frameworks were analysed through the lenses of CARE-KNOW-DO. Our Padlet generated 60 URLs shared by participants about initiatives relevant to foster CCSE and open schooling, for example, learning materials, open educational project, portals about climate change in schools, articles about sustainability and references about green skills (collected through an online noticeboard: https://padlet.com/alexandraokada/CCSE accessed on 15 November 2022).

In this question, we focused on examining skills that could help learners to build competences. We examined the interrelationships between the latest green skills components, considering values to be cared about, knowledge to be known, and attitudes towards what is to be done, putting knowledge and values into action. The CARE-KNOW-DO model of green skills (Figure 3) is underpinned by ten of the latest documents collected in our Padlet (Department of Education—GOV UK, 2022 [2], 2022 [4]; Bianchi et al., 2022 [17]; Deloitte, 2022 [18]; Alliance, 2022 [19]; Taskforce, 2021 [20]; British Academy, 2021 [21]; Patterson et al., 2022 [55], Kwauk & Casey, 2022 [56], Malagrida et al. [57]).

The structure of these lenses was inspired by Kwauk & Casey's policy-focused work [56], which provided a model for presenting three groups of skills for Green Jobs, Green Transformation and Green Life. In our model, focused on education, (Figure 4), we positioned these three groups to connect formal, informal and non-formal learning among schools, universities, enterprises and communities, a process which can be enhanced by open schooling. These macro lenses are at the left, skill for green careers; at the centre, skills for green life; and, at the right, skills for a green society. The CARE-KNOW-DO provided a

micro-filter to group and link the skills, aiming at supporting educators, communities, and organisations. See also the dataset (URL: https://7a0a0732.flowpaper.com/ccse/ accessed on 15 November 2022) which illustrates the framework applied to present resources.

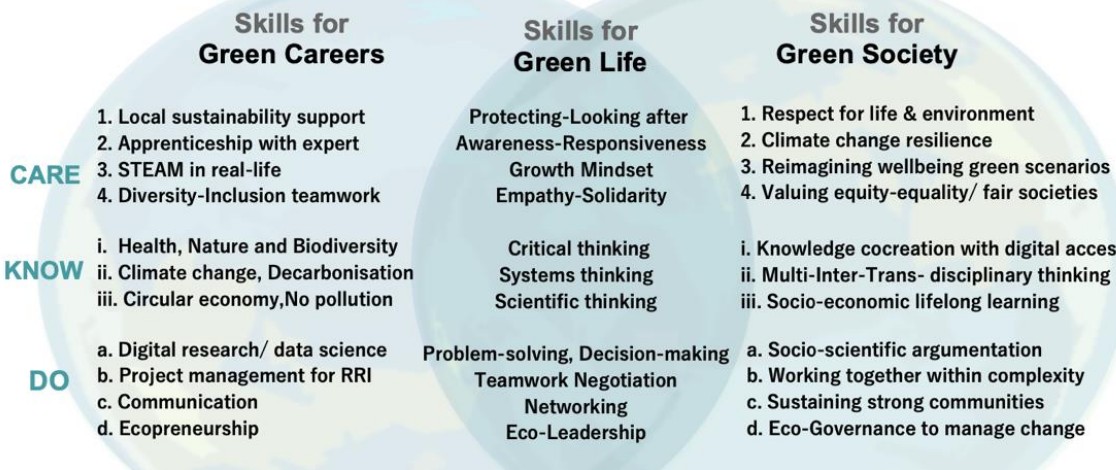

**Figure 4.** Skills for Green Careers, Green Society and Green Life supported by CARE-KNOW-DO open schooling framework (Prepared by the authors).

Firstly, 'CARE' refers to the type of skills learners need to develop to ensure that values and local needs are taken into account from the perspective of enterprises (green careers), individuals (green life), and communities (green society). In terms of Green Careers, four skills were selected:

1. **Local sustainability support**: ability to meet local demands, adapt to local settings, and become actively involved in local sustainable development.
2. **Apprenticeship with experts**: ability to learn from scientists or professionals about their green jobs and become aware of the green skills that may be needed in new kinds of careers.
3. **STEAM in real-life**: ability to identify real-life issues using Science, Technology, Engineering, Arts, and Maths relevant for society.
4. **Diversity-Inclusion teamwork:** ability to work in inclusive and diverse teams with underrepresented or underserved people.

Regarding Green Societies, four skills are relevant, underpinned by inclusion, diversity, equity, and social-economic-environmental justice:

1. **Respect for life and environment**: ability to value health and wellbeing and reflect on socio-scientific issues that may affect living organisms and the environment.
2. **Climate change resilience**: ability to identify uncertain and unexpected needs caused by climate change and discuss how to respond to minimise the effects on sustainable development.
3. **Reimagining wellbeing green scenarios**: ability to foster green urban spaces, rural areas and preserve green cultural heritage.
4. **Valuing Equity-equality/fair societies:** ability to contribute to teamwork, value equality and support fair communities. (e.g., in group discussion, teamwork, community-based projects).

The intersection of both lenses is sustained by skills for Green Life at the centre, which refers to individuals' behaviours and actions as the agents of change towards sustainable living:

1. **Protecting-looking after**: ability to take care of oneself and others.

2.  **Awareness-Responsiveness**: ability to recognise issues and respond to them promptly.
3.  **Growth mindset**: ability to cope with emotions, cognitive challenges, and uncertainty.
4.  **Empathy-Solidarity:** ability to support others with understanding and kindness towards sustainability.

Secondly, 'KNOW' refers to transferable and subject matter knowledge skills to solve problems, develop new products or processes, or train others, which are clustered into three domains.

In terms of Green Careers, four skills were selected:

i.  **Health, nature, and biodiversity:** ability to understand about good health and life on Earth, freshwater, renewable energy, nutrition and food security, medicines, and protection of living beings.
ii.  **Climate change decarbonisation:** ability to understand about global warming and greenhouse gas emission, mitigation actions to reduce emissions and adaptation action to manage the risks.
iii.  **Circular economy with no pollution**: ability to understand about regenerating nature and keeping waste out of the environment by sharing, reusing, repairing, and recycling existing materials and products as long as possible.

Regarding Green Societies, three skills were identified:

i.  **Knowledge cocreation with digital access**: ability to communicate one's own understanding and evaluate others' views to build knowledge collaboratively on the internet.
ii.  **'Multi/inter/trans' disciplinary thinking**: ability to participate in multi-, inter- or trans- disciplinary learning incorporating integrated thinking, supported by all branches of sciences (natural sciences, mathematics, human and social sciences).
iii.  **Socio-economic lifelong learning**: ability to benefit from lifelong learning for future well-being, supporting green socio-economy.

The intersection between both groups represents the central cluster of key skills for green life:

i.  **Critical thinking**: ability to analyse, evaluate and develop an argument, along with the evaluation of evidence, including concepts, principles, and assumptions.
ii.  **Systems thinking**: ability to explore and develop effective strategies in complex contexts considering the whole and the relationships of interconnected components instead of splitting it down into its parts.
iii.  **Scientific thinking**: ability to solve problems supported by scientific sources, scientific methods, and scientific experts.

Thirdly, 'DO' brings together three sets of four skills for learners to establish in action towards green careers:

a.  **Digital research and data science:** ability to find an answer to a question or a solution to a problem with a methodical investigation including data analysis supported by technologies.
b.  **Project management for RRI**: ability to create and develop projects with the team members considering scientific and technological advances aligned with societal needs.
c.  **Communication**: ability to communicate clearly and scientifically considering diversity, inclusion, and equity.
d.  **Ecopreneurship**: ability to situate nature and climate at the core of a business model with income for local livelihoods supported by more resilient, inclusive, and sustainable approaches.

Congruently, skills to enhance green societies are:

a.  **Socio-scientific argumentation**: ability to discuss the social consequences of science underpinned by arguments and evidence from multiple, reliable sources.
b.  **Working together within Complexity**: ability to investigate complex questions, multiple scenarios, weigh the evidence, identify, and interpret uncertainty.

c.   **Sustaining strong communities:** ability to identify opportunities and challenges to develop and sustain strong communities by engaging all members and external public.

d.   **Multi-level governance to manage change**: ability to design, implement and evaluate policies, in order to support positive change by facing barriers, maximizing benefits and reducing risks.

The intersection of both groups is supported by key skills for green life:

a.   **Problem-solving/Decision-making**: ability to detect misinformation and disinformation as well as identifying and using timely, trustworthy, and accessible sources of information to respond to issues, problems and evaluate options, decision-making and take action, the need for decisions.

b.   **Teamwork Negotiation:** ability to support group negotiation that can lead to improved outcomes, resolution of conflicts and increased team cooperation.

c.   **Social Networking:** ability to build and nurture interactions, connections, and relationships with others.

d.   **Eco Leadership:** ability to lead collective actions supported by a community to ensure a healthy society and healthy environment with sustainable development.

## 5. Discussion and Final Remarks

In this article, we set out the results from a Delphi study into the state-of-the-art in climate change and sustainability education (CCSE) and related topics, such as carbon literacy. Our findings highlight make a case for a 'New Education Movement', which combines the best elements of radical educational thought from the past, with the increasing urgency of change required by the current crises affecting the globe.

We do this from a background of involvement in (largely) EU-funded projects addressing the relationship of STEAM education to society, under various banners such as Inquiry-Based Learning, Responsible Research and Innovation, and Open Schooling. It is remarkable that some of these concepts have been around since the late 19th century [45], and equally remarkable that it has taken until now for many of the principles of this movement to be accepted by the educational mainstream, albeit grudgingly in some cases. However, we think it is helpful to use the idea of a 'movement' to bring together the diverse strands of expertise represented in the study. This is partly as an antidote to the idea of the 'project', which in the terms of the European Commission and other funding bodies, is time limited and exclusive to project members in many respects. We will return to this topic at the end of the article, but first we will explore the results of the study itself.

### 5.1. What Do We Know from the Participant Contributions?

The combined findings from contributors to this study suggest that changes to current teaching and learning practices in CCSE, in the UK and elsewhere, can come from within the system rather than from tinkering with policy. In countries whose current policy supports the overall aims of CCSE, the task of educators with local and national policy makers should be to ensure that the key strategies are actually applied to practice.

We draw three main conclusions that support this overall view:

- There is a powerful ecosystem of organisations and individuals—to be connected as learning ecologies [58]—willing to support climate change and sustainability education (CCSE).
- There are sufficient resources and materials already in existence.
- There is a lack of coordination across the ecosystems, which prevents CCSE from being fully implemented.

The changes we have identified as being necessary, in the opinion of contributors and with a degree of hindsight, could constitute a New Education Movement for the contemporary crisis. We use the term 'movement' to distinguish it from 'strategies', 'projects' or 'initiatives', although these are necessary to support change, as are communities of practice dedicated to the movement.

We therefore make overall recommendations at three levels:

- Local: adoption of CCSE scenarios for practical action, to be embedded in the existing curriculum, thus requiring few changes, and reflecting what is already happening on the ground.
- National: DfE policy agenda supported through evidence-based outcomes provided by our movement and originating from the forum.
- Global: formation of coalition/alliances at international level—Climate Action brings opportunity for grants, proposals, networking, and collaboration.

The main task for researchers in this field, and the authors, is to maintain the momentum of the forum by setting concrete targets and carrying out specific actions. These should initially relate to the local and national levels. For example, curating a page on the website of an ongoing project such as CONNECT, or an organisation such as the Association for Science Education in the UK, might assist members of the forum to disseminate their materials more widely.

At the national level, there could be some independent evaluation of progress towards the Strategy and adoption of the Action Plan as mentioned above, in order to hold the Government to account and to shape future policy. Most, if not all, participants have ongoing links to schools and could easily survey opinion as to whether the strategy is working. It would also be possible to work with 'sustainability leads' as described in the strategy, perhaps by pro-actively organising a conference in 2023/24 to bring them together under the auspices of the forum.

In terms of the global aspect, there is already a considerable amount of relevant activity across Europe and beyond that is building bridges between the bio-based industry and the education system by interlinking universities, innovation labs, and Research and Innovation centres with industrial actors and regions [59,60], for example, Biobec.eu and EDBioEC projects. (See ORDO Padlet data—open access (link)).

### 5.2. What Are Our Recommendations for Further Actions Including Collaboration between Education Providers to Enhance CSSE?

Partners' expertise for knowledge exchange covered a variety of topics in which participants were involved, either as leaders or practitioners. A strategy to achieve maximum sustainability should be a blend of top-down and bottom-up approaches by which local authorities, NGOs and universities work together with government, schools, and individuals [60,61]. Three themes emerged. The first theme concerns teacher training for green education including initial teacher education programmes, continuing professional development, and lifelong learning, climate change and environmental education workshops. The second theme focused on the curriculum for green education, bridging formal and non-formal learning, and including learning programmes on environmental education, education technology and resources for learning, curriculum design, schools' best practices, organisational projects, and practitioners' initiatives. The third theme focused on research-based policy, which included policy research and reports, climate action leadership programmes, climate education research and responsible research and innovation.

### 5.3. What Kind of Activities Might We Need in Each Phase of the Framework?

The approach we are suggesting here requires changes at all levels of the education system to ensure high-quality CCSE.

CARE, as the basis for education, needs to be actively taught, not just as an underlying, bureaucratic assumption of the system (as in 'duty of care'), but also as a philosophical principle. It undoubtedly involves a more student-centred approach, but one where individual and collective development of the student group is balanced and mutually reinforcing, through interaction.

This means, firstly, that assessment needs to become much more sophisticated. Realistically, the existing system is so deeply embedded in the educational 'sea of consciousness' that it cannot simply be discarded. Rather, we need to supplement examination and qualifi-

cation systems with more subtle methods that are truly formative, perhaps in a return to the idea of *Bildung, (Bildung refers to the German tradition of self-maturation and self-cultivation, which means a process of personal and cultural maturation)*, the development of individuals as members of society. The idea of a career needs to evolve into the idea of a fulfilling life, in which individual and collective needs are in a harmonious balance.

Secondly, we need to let go of the idea of a fixed curriculum; however much it might be underpinned by fundamental 'big ideas' as the foundation needed for students to progress. Perhaps a useful metaphor is that of the playlist. Spotify has ten million songs or more, but also has millions of individually created playlists which reflect the preferences and needs of those creating them. We have, potentially, the digital resources to do this for knowledge as well as music. Open badges and similar methods provide the means to compile individual and collective lists of what is known and what is understood. More than simply listing 'things known', this can be combined with a narrative approach in which students record their own stories, as evolving chapters in a life, accounts of where they stand and what they understand about the world.

Thirdly, these changes need to be supported by an open pedagogy in which knowledge flows across all boundaries, not just from teacher to student, but from student to student and from the world to all learners. Despite the embedded disdain of many educators towards informal sources such as YouTube and Wikipedia, we need to recognise that students already benefit from the use of such sources in their personal journeys. Some of these uses might seem mundane. I recently used YouTube to learn how to fix a toilet flushing system. This does not make me a plumber, but it broadens the definition of plumbing and makes it part of my identity in a small way. Fixing dual flush panels is on my playlist.

We need to recognise this, not to regulate open learning provision, which is futile, but to empower students to take learning opportunities, to be critical and discerning of resources, and to share what they know with others for the common good. We also need to begin mapping care in the education system. Where is it showing up? How much are students influenced in becoming carers—independent thinkers and agents of change, in a wide sense? How are pedagogical and personal relationships working to promote care?

*5.4. Why Should We 'CARE, KNOW and DO'?*

We need to introduce provocations that are valuable and matter to students. Most of the ideas for current scenario activities or projects are based on ideas from science-related fields, for example 'rewilding' of rural areas. However, many of the changes that will become necessary are subtractive in relation to our current lifestyles. In other words, we simply have to stop doing certain things. It would be provocative to suggest, for example, that the use of motorcycles be limited to essential travel or deliveries. The motorcycle business is a significant one in Europe and elsewhere [62] but the impact of motorcycles is not just their carbon emissions but also their acoustic impact in both cities and rural areas. Yet leisure motorcyclists are notoriously uncaring about their impact and fiercely defensive of their sport. Rather than invite a scientist to discuss this topic, invite a biker! This brings us to the topic of resistance [63].

As with all arguments based on normativity, there is a danger that a New Education Movement based on climate change and related crises could provoke resistance, a desire to defend one's current position and privileges for as long as possible. This might also be seen as an intergenerational issue and is the inverse of the historical idea that children should enjoy better prospects and lifestyles than their parents. Most of today's students face a future that looks increasingly bleak. Here, we suggest that the 'DO' component of the framework is the way forward. Students need to be supported in taking action, even when such actions conflict with traditional school behaviour, such as climate strikes or demonstrations. Students need respect, whether this is from teachers, politicians, or peers. Action breeds respect, so long as it is embedded in an ethical framework and justified by

knowledge. The three components of the framework CARE-KNOW-DO towards learning ecologies (Figure 5) are thus mutually supportive and equally essential.

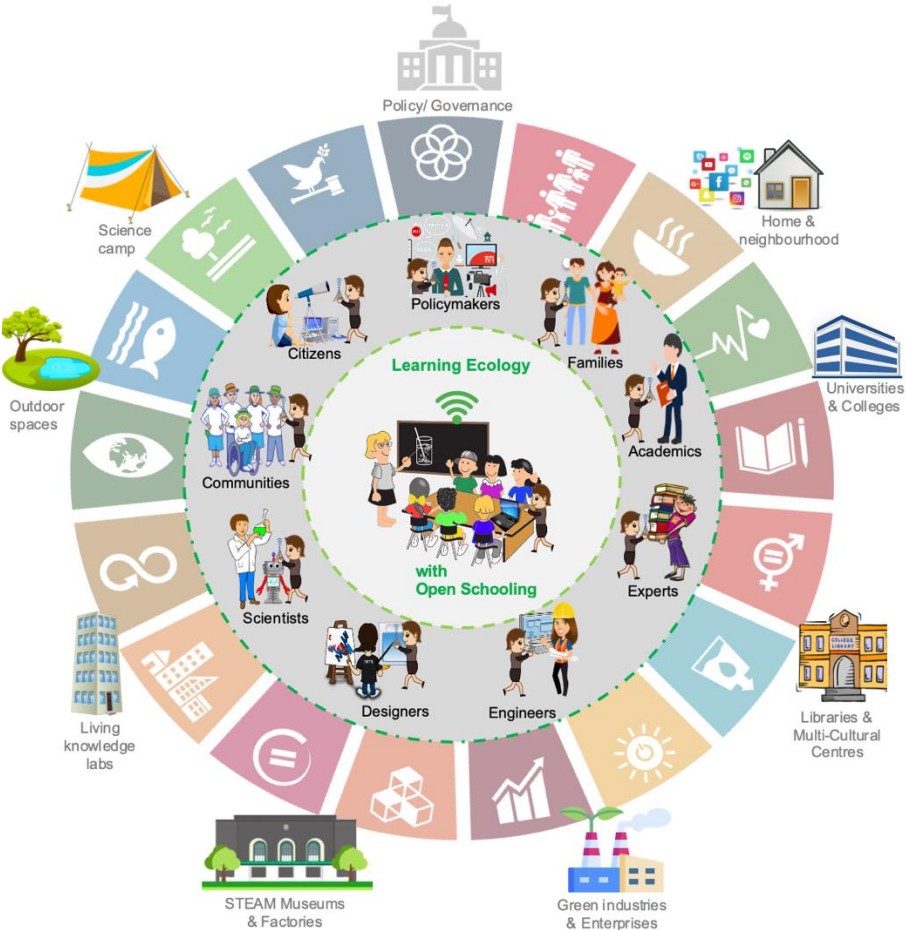

**Figure 5.** CCSE movement of Learning Ecologies with networks, open schooling, and the CARE-KNOW-DO framework.

The term "learning ecologies" refers to ecological perspectives on human development [64], whose processes and outcomes occur within a wide range of rich habitats. This means that the specific learning ecologies inhabited by students shape their learning opportunities. The more diverse, interactive, challenging and engaging their learning ecologies are, the richer, more meaningful, and more transformative their learning will be. We argue that learning ecologies can be built through open schooling networks, thus providing students with a rich diversity of habitats and encounters with distinctive societal actors. These habitats include schools, outdoor spaces, living knowledge labs, and universities, whilst actors include families, local communities, educators, and policy makers at local, regional, and national levels.

These interactions, in conjunction with real-life problem solving, will expand students' capabilities to CARE about, KNOW about and DO something about, in order to ensure sustainability. The CCSE movement supported by open schooling and the CARE-KNOW-DO framework is, therefore, an ideal vehicle to nurture and enhance students' affective, mental, and physical development.

This study explored the literature gap related to integrating and scaling up green initiatives to provide the necessary impact for change. The original contribution to knowledge of this paper is an integrated approach to fostering Climate change and Sustainability Education (CCSE) movement of learning ecologies that connect practitioner networks, open schooling, and the CARE-KNOW-DO framework for engaging and meaningful STEAM

education. The study provides recommendations for CCSE practice, policy, and research relevant to developing a green curriculum and green skills in school systems, supported by a wide range of non-formal initiatives.

Localized, relevant, and appropriate policy development [65,66], will be fundamental to inform future global sustainability education initiatives [67]. For the next stage of the movement, therefore, we suggest the use of the expanded **CARE-KNOW-DO** framework as an analytical tool to examine current educational practices, and to provide an outline map for the way forward.

**Author Contributions:** Both authors were responsible for conceptualization, methodology, instruments, validation, investigation, and writing. A.O. was responsible for data curation and analysis, first draft preparation, and final typesetting, prepared all images, supervision, project administration, funding acquisition. P.G. contributed with Delphi facilitation for data generation, discussion, recommendations, and review. All authors have read and agreed to the published version of the manuscript.

**Funding:** This Delphi study is funded by The Open University UK. It is also funded by the European Union part of the research programme Climate Change and Sustainability Education developed by the projects: ENGAGE—Equipping the next generation for active engagement in science n. 612269 and CONNECT—inclusive open schooling with engaging and future oriented science n. 872814.

**Institutional Review Board Statement:** The study was conducted in accordance with the Declaration of Helsinki, and approved by the Ethics Committee of The Open University UK-HREC/3825.

**Informed Consent Statement:** Informed consent was obtained from participants involved in the study.

**Data Availability Statement:** The open database can be accessed, downloaded, and reused: Okada and Gray (2023) The GREEN FORUM COLLECTION about Climate Change and Sustainability Education (CCSE) with green initiatives, green curriculum recommendations, and green skills CCBYSA. Open Research Data Online. The Open University. https://ordo.open.ac.uk/account/home#/collections/63 24032 https://doi.org/10.21954/ou.rd.21836490, (URLs accessed on 15 November 2022).

**Acknowledgments:** We are grateful to all members of the Green Forum network who contributed to this study including to the catalogue of green initiatives that can be accessed through this URL (https://7a0a0732.flowpaper.com/ccse/, accessed on 15 November 2022) Thanks to our colleagues and practitioners and reviewers, including J.Sanders (CCSE teachers' educator), G. Young and T. Sherborne (CCSE curriculum developer) for providing useful feedback.

**Conflicts of Interest:** The authors declare no conflict of interest.

### Abbreviation

| | |
|---|---|
| CCSE | Climate Change and Sustainability Education |
| WHO | World Health Organisation |
| Ofsted | Office for Standards in Education, Children's Services and Skills |
| DfE | Department for Education in the UK |
| STEAM | Science, Technology, Engineering, Arts, Mathematics |
| RRI | Responsible Research and Innovation |
| RSE | Royal Society of Edinburgh |

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
