# Peer review of "A Climate Change and Sustainability Education Movement: Networks, Open Schooling, and the ‘CARE-KNOW-DO’ Framework"

_sustainability, doi:10.3390/su15032356_

Round 1

Reviewer 1 Report

The manuscript is focused on the interplay and close cooperation between universities, schools, enterprises, and wider society for the joint development of actions for CCSE ‘Climate Change and Sustainability Education’. The manuscript has got an empirical character. As it is possible to observe, the manuscript has got qualitative approach toward obtaining and analyzing of data. The chapters are presented in the little bit interesting format, but after careful reading this ranking and presentation of chapters is logical. On the first look, it seems relatively long, but all kinds of information presented in the text are important for the holistic understanding of the text and ideas of the study. The text is written in understandable form. I have got only comments of minor character.

1. Please add minimally one sentence into the abstract. The abstract should include kinds of information from every parts of the manuscript. So please add minimally one sentence from the theoretical part.

2. The theoretical part of the manuscript is relatively long and some parts are written in little bit epic form. Are you convinced, that all kinds of information are necessary for the manuscript as the whole unit? Maybe some of them, would be eliminated or minimally shortened.

3. The kinds of information about participants are written in very brief form. For this kind of study it very important to provide kinds of information for the understanding of the selection of participants. Because the abilities and qualities of participants could influence the overall findings and also the answers on research questions. So please add more detailed kinds of information about participants of the study.

4. Other parts of the manuscript are written on high level, I have not got other comments, only revise references according guidelines for authors.

I hope my comments are helpful.

Author Response

Reviewer 1

Please add minimally one sentence into the abstract. The abstract should include kinds of information from every parts of the manuscript. So please add minimally one sentence from the theoretical part.

Done.

Underpinned by the CARE-KNOW-DO theoretical principles (Okada, 2022) we explore CCSE issues and provide a foundation for a new education movement to combine strategies, initiatives, and interventions.

2. The theoretical part of the manuscript is relatively long and some parts are written in little bit epic form. Are you convinced, that all kinds of information are necessary for the manuscript as the whole unit? Maybe some of them, would be eliminated or minimally shortened.

We agreed with the 2nd reviewer to keep the content to make the concepts and foundation clear

3. The kinds of information about participants are written in very brief form. For this kind of study it very important to provide kinds of information for the understanding of the selection of participants. Because the abilities and qualities of participants could influence the overall findings and also the answers on research questions. So please add more detailed kinds of information about participants of the study.

Done.

All members of the green forum received an invitation to attend an online workshop “Education Policy and Climate cHange (EPOCH) - Building networks across boundaries”. They were informed that the aim of this event was to enable policymakers to learn from each other, supported by Delphi principles and procedures including iteration, reflection, and anonymity. They were genuinely committed to Sustainability Education and Climate Change and joined the event based on their availability and willingness to volunteer.

4. Other parts of the manuscript are written on high level, I have not got other comments, only revise references according guidelines for authors.

Done.

Reviewer 2 Report

This article is relevant in claiming attention to a new path in education, both regarding contents and methods. I just suggest to move paragraph 2.3 (lines 312-323) to methodology. Take care also of some citation that is not following the general indications (lines 59, 62, 246, 300).

Author Response

Reviewer 2

This article is relevant in claiming attention to a new path in education, both regarding contents and methods. I just suggest to move paragraph 2.3 (lines 312-323) to methodology. Take care also of some citation that is not following the general indications (lines 59, 62, 246, 300).

Done.

Reviewer 3 Report

The authors present a UK- based and funded program around climate change and sustainability education and the Care - know- do framework. This was a long but very interesting read. Thank you for laying out the framework and give a wealth of quotes that make the text very readable and understandable

I'd like to ask the authors to consider the following modifications (all minor):

Line 47 - define zero emission , emission of what?

spacing issues in lines 50-56

Line 108: 

After introducing the term open schooling add a sentence how it is different from other approaches like citizen science, community science, participatory research. Is open school a school based mentorship model or not? If yes, who is mentoring?

Line 115 - needs a reference

Line 134 - add "motivated" (i think goal is to make students competent and motivated?)

Line 137 - spacing issue

Line 155 - I assume this statement is true for western societies both other societies and cultures exist where environmental stewardship is critical and instilled at a very early age

Line 186: Include details. What does sufficient mean? # of transactions? Quality of transaction? Who decides? Students/teachers/admins? Is student voice an element here or not?  

Line 205: This is more of a comment: I'm surprized that there is no step in between personal to global level. "I care about the future of the local environment" would make sense for me. 

Line 213: There is also the idea that school attempts to train for jobs that don't exist yet, and we do not know what skills will be needed for those jobs. 

Minor spacing issues throughout paragraph. 

Line 354: Define EPOCH, you do it in line 371, so just move it up. 

Line 364: Were members paid to participate, what was the incentive overall? What was the overall recruitment strategy to select the 50 members?

Line 408 - spacing issues

Line 448 - define and clarify Ofsed

Line 451 -  and clarify RSE 

Line 473 - what is DfE?

Line 510 - reference for NVivo needed. 

Line 514 - OK looking at the analysis two other words stick out - climate and education. Start with those first, then loop into the curriculum. 

Line 629 - modify to COVID-19 pandemic. 

Line 645 - I kindly ask to include other recent events that had a major impact on UK's socio-economic reality - "Brexit" or the UK leaving the European Union. I think this cannot be ignored. 

Line 658 ongoing - some spacing issues

Line 683 - capitalize Zoom, check if a trademark sign is needed and add a reference

Line 722 - age of pupils missing?

Line 745 - define OFSTED. 

Line 757 - spacing issue

Line 813 - spacing issue

Line 816 - Thank you for this closing statement. Add that the most vulnerable parts are highly affected by climate change and climate injustice. 

Line 858 - define STEM

Line 930 - spacing issues

Line 941 - add "for this community of practice" 

Line 994 - define Bildung earlier and keep spelling consistent. 

Line 1021 - carers = stewards?

Author Response

Reviewer 3

The authors present a UK- based and funded program around climate change and sustainability education and the Care - know- do framework. This was a long but very interesting read. Thank you for laying out the framework and give a wealth of quotes that make the text very readable and understandable

Done.

Line 47 - define zero emission , emission of what?

spacing issues in lines 50-56

Done.

In the UK, for example, the new ‘Green Jobs Taskforce’ was launched in 2020 to support the creation of two million green jobs and reach ‘net zero’. This refers to the transition to a low carbon economy with a high-skill job market capable to keep the balance between the production and removal of the same amount of greenhouse gas from the atmosphere by 2050

Line 108: 

After introducing the term open schooling add a sentence how it is different from other approaches like citizen science, community science, participatory research. Is open school a school based mentorship model or not? If yes, who is mentoring?

Done.

Open schooling can be implemented through various approaches,  enabling students to solve problems or questions supported by families and scientists. In this respect it is similar to for example, citizen-science, community-based participatory research, consensus conference, cocreation, design thinking, inquiry-mapping, system-oriented dialogue model and collaborative project based learning [63]. As the name suggests, open schooling operationalises these concepts in primary and secondary education, through local partnerships.

Line 115 - needs a reference

Line 134 - add "motivated" (i think goal is to make students competent and motivated?)

Line 137 - spacing issue

Done.

This is all relevant for students’ affective engagement, and to raise the intrinsic motivation necessary for them to develop agency

Line 155 - I assume this statement is true for western societies both other societies and cultures exist where environmental stewardship is critical and instilled at a very early age

Done.

Developing the widest possible range of green competences, starting from an early age, is essential for such active engagement. 

Line 186: Include details. What does sufficient mean? # of transactions? Quality of transaction? Who decides? Students/teachers/admins? Is student voice an element here or not?  

Done.

In Open Schooling, students experience sufficient transactions to connect what they learn in school with the world outside the classroom, in order to develop judgement, empathy, and care for the future. Care has two common usages, and it is important to harness both if we want to improve climate change education. It means firstly ‘looking after’ the environment and life around us, as stewards for future generations. Secondly, it also means something that matters or has value to me, such as, for example, eliminating pollution, poverty, and global warming. Care helps students, in conjunction with experts and society, to develop the ethical values and responsible attitudes that are fundamental to Responsible Research and Innovation (RRI).

Line 205: This is more of a comment: I'm surprized that there is no step in between personal to global level. "I care about the future of the local environment" would make sense for me. 

Done.

●      I am in education because I care, and others care about me

●      I care about myself and my future

●      I care about the future of others close to me and my local environment

●      I care about the future of others and their local environments

●      I care about the future of the planet as a global ecosystem

Line 213: There is also the idea that school attempts to train for jobs that don't exist yet, and we do not know what skills will be needed for those jobs. 

Minor spacing issues throughout paragraph. 

Done.

Learning how to learn will be necessary for jobs and skills that do not yet exist.

Line 354: Define EPOCH, you do it in line 371, so just move it up. 

Done.

Line 364: Were members paid to participate, what was the incentive overall? What was the overall recruitment strategy to select the 50 members?

Line 408 - spacing issues

Line 448 - define and clarify Ofsed

Line 451 -  and clarify RSE 

Line 473 - what is DfE?

Line 510 - reference for NVivo needed.

Done.

All members of the green forum received an invitation to attend an online workshop “Education Policy and Climate cHange (EPOCH) - Building networks across boundaries”. They were informed that the aim of this event was to enable policymakers to learn from each other, supported by Delphi principles and procedures including iteration, reflection, and anonymity. They were genuinely committed to Sustainability Education and Climate Change and joined the event based on their availability and willingness to volunteer.

Line 514 - OK looking at the analysis two other words stick out - climate and education. Start with those first, then loop into the curriculum. 

Line 629 - modify to COVID-19 pandemic. 

Line 645 - I kindly ask to include other recent events that had a major impact on UK's socio-economic reality - "Brexit" or the UK leaving the European Union. I think this cannot be ignored. 

Done.

Climate and Education were the key words most mentioned by participants. These were followed by Curriculum, which was the key concept that emerged in both sessions.

Recent events such as Brexit (the withdrawal of the United Kingdom from the European Union), the invasion of Ukraine, economic crisis and poverty growth, or the scandal over the export of UK plastic waste to Turkey, underline the need for a higher level of climate consciousness at all levels up to world leadership.

Line 658 ongoing - some spacing issues

Line 683 - capitalize Zoom, check if a trademark sign is needed and add a reference

Line 722 - age of pupils missing?

Line 745 - define OFSTED. 

Line 757 - spacing issue

Line 813 - spacing issue

Done.

The second recommendation towards an interdisciplinary curriculum for all ages (5-25) and educational levels (from primary to higher education), supported by teacher professional development, is to encourage and prepare teachers to implement cross-curricular practices.

The second recommendation towards an interdisciplinary curriculum for all ages (5-25) and educational levels (from primary school to higher education), supported by teacher professional development, is to encourage and prepare teachers to implement cross-curricular practices

Acronyms added.

Line 816 - Thank you for this closing statement. Add that the most vulnerable parts are highly affected by climate change and climate injustice. 

Line 858 - define STEM

Line 930 - spacing issues

Line 941 - add "for this community of practice" 

Line 994 - define Bildung earlier and keep spelling consistent. 

Line 1021 - carers = stewards?

Done.

We are using now STEAM (also described in the acronyms)

. We use the term ‘movement’ to distinguish it from ‘strategies’, ‘projects’ or ‘initiatives’, although these are necessary to support change, as are communities of practice dedicated to the movement.

Bildung (a German word meaning something like self-maturation)

in becoming carers – independent thinkers and agents of change,
